# Wave-controlled aliasing in parallel imaging (Wave-CAIPI): Accelerating speed for the MRI-based diagnosis of enhancing intracranial lesions compared to magnetization-prepared gradient echo

**Hyunji Oh, Younghee Yim** ⦿*, **Mi Sun Chung, Jun Soo Byun**

Chung-Ang University Hospital, Chung-Ang University College of Medicine, Seoul, Republic of Korea

* youngheeyim@gmail.com

**Data Availability Statement:** All relevant data are within the paper and its Supporting Information files.

## Abstract

### Purpose

We aimed to validate the diagnostic performance of accelerated post-contrast magnetization-prepared rapid gradient-echo (MPRAGE) using wave–controlled aliasing in parallel imaging (Wave-CAIPI) for enhancing intracranial lesions, compared with conventional MPRAGE.

### Methods

A total of 233 consecutive patients who underwent post-contrast Wave-CAIPI and conventional MPRAGE (scan time: 2 min 39 s vs. 4 min 30 s) were retrospectively evaluated. Two radiologists independently assessed whole images for the presence and diagnosis of enhancing lesions. The diagnostic performance for non-enhancing lesions, quantitative parameters (diameter of enhancing lesions, signal-to-noise ratio [SNR], contrast-to-noise ratio [CNR], and contrast rate), qualitative parameters (grey-white matter differentiation and conspicuity of enhancing lesions), and image qualities (overall image quality and motion artifacts) were also surveyed. The weighted kappa and percent agreement were used to evaluate the diagnostic agreement between the two sequences.

### Results

Wave-CAIPI MPRAGE achieved significantly high agreement for the detection (98.7%[460/466], κ = 0.965) and diagnosis (97.8%[455/466], κ = 0.955) of enhancing intracranial lesions with conventional MPRAGE in pooled analysis. Detection and diagnosis of non-enhancing lesions (97.6% and 96.9% agreement), and diameter of enhancing lesions (P>0.05) also demonstrated high agreements between two sequences. Although Wave-CAIPI MPRAGE show lower SNR (P<0.01) than conventional MRAGE, it fulfilled comparable CNR (P = 0.486) and higher contrast rate (P<0.01). The qualitative parameters show similar value (P>0.05). The overall image quality was slightly poor, however, motion artifacts were better in Wave-CAIPI MPRAGE (both P = 0.005).

**Funding:** YY got grant for this research. This research was supported by the Chung-Ang University Research Grants in 2021. There was no additional external funding received for this study.

**Competing interests:** The authors have declared that no competing interests exist.

## Conclusion

Wave-CAIPI MPRAGE provides reliable diagnostic performance for enhancing intracranial lesions within half of the scan time compared with conventional MPRAGE.

## Introduction

T1-weighted magnetization-prepared rapid gradient echo (MPRAGE) technique has been widely used for structural imaging due to gray-white matter contrast and superior image quality [1]. In clinical settings, MPRAGE enables multi-directional reconstruction imaging and serves as a precise anatomical reference for advanced MRI data and brain volumetric analysis [2, 3]. Moreover, with contrast enhancement, MPRAGE plays an essential role in the detection and evaluation of metastases and gliomas and is recommended by the guidelines such as the Response Assessment in Neuro-Oncology (RANO) criteria [4–6].

Both in research and clinical settings, MPRAGE usually requires a relatively long scan time to achieve high spatial resolution and proper T1 weighted contrast in need of long inversion time [1]. Therefore, MPRAGE is generally applied with various parallel acquisition techniques to resolve the prolonged scan time issues [1]. The parallel acquisition technique is a k-space undersampling method that decreases the number of phase-encoding steps through the use of coil sensitivity encoding from a multichannel receiver array [7–9]. Nevertheless, increasing number of parallel acquisition factors is limited, because the image quality is ruined by noise amplification from the larger geometric (g) factor and the acquisition of fewer data points when the parallel factor increases.

The wave-Controlled Aliasing In Parallel Imaging (CAIPI) is a recently proposed parallel imaging technique for higher accelerated MR image acquisition in daily practice [10, 11]. Wave-CAIPI was developed by combining the bunched phase encoding and 2D-CAIPI to generate sinusoidal Gy and Gz gradients with a π/2 phase shift between the waveforms and it creates a corkscrew 3D trajectory in k-space to achieve controlled aliasing in all three spatial directions (x, y, z) [10]. Therefore, Wave-CAIPI has taken advantage of full 3D coil sensitivity information and enables highly accelerated volumetric imaging with negligible g-factor penalty and low artifacts, including MPRAGE [10, 11]. Recently, precontrast Wave-CAIPI MPRAGE has been validated in volumetric analysis including patients with dementia using 20- and 32-channel coils [12, 13]. Also, recent study demonstrated that fast scan using contrast-enhanced Wave-CAIPI 3D T1-MPRAGE was noninferior to the 3D T1-MPRAGE sequence in visualizing and diagnosing enhancing brain lesions [14].

Based on these results, we hypothesized that Wave-CAIPI MPRAGE could have sufficient spatial resolution for the detection of intracranial lesions in post-contrast scans for clinical use.

Therefore, we aimed to compare the diagnoses of enhanced intracranial lesions between conventional enhanced 3D T1WI MPRAGE without Wave-CAIPI acceleration and Wave-CAIPI MPRAGE. We also compared the diagnostic performance of non-enhancing intracranial lesions and the values of quantitative and qualitative parameters in both sequences.

## Materials and methods

This retrospective study was approved by the institutional review board of Chung-Ang University Hospital (IRB number: 2006-027-19320), and informed consent was waived owing to the retrospective study design by the institutional review board of Chung-Ang University Hospital (IRB number: 2006-027-19320). We reported methods and results according to the

STROBE (strengthening the reporting of observational studies in epidemiology) guidelines [15].

## Study population

We retrospectively assessed consecutive patients who underwent contrast-enhanced brain MRI examination at a single tertiary center between July 2018 and September 2018. The inclusion criteria of this study included the following: (a) patients who underwent contrast-enhanced brain MRI with both Wave-CAIPI and conventional MPRAGE sequences, (b) adults (age > 20 years), and c) patients without any contraindication to MRI or contrast enhancement. The exclusion criteria included the following: (a) severe motion or metal artifact and (b) data reconstruction failure. A total of 233 patients were enrolled in this study. We also retrospectively collected demographic data by reviewing electronic medical records, including age and sex.

## Image acquisition

All studies were performed with a two 3-T MR imaging system (Magnetom Skyra, SIEMENS, Erlangen, Germany) using 64-channel head coils in the IDEA environment. Gadobutrol (Gadovist; Bayer Healthcare, Berlin, Germany) was injected intravenously at 0.1 mL/kg using a 3-way stopcock. Post-contrast MR scanning was executed just after the injection of contrast media in the following order: Wave-CAIPI MPRAGE → conventional MPRAGE.

The detailed MR scan parameters of the enhanced Wave-CAIPI MPRAGE and conventional MPRAGE in this study are shown in Table 1. The total acquisition times were 4 min 30 s for conventional MPRAGE and 2 min 39 s for Wave-CAIPI MPRAGE, achieving a 42% reduction in scan time.

## Image analysis

Two radiologists (H.O. and M.S.C. with 2 years and 9 years of experience in brain imaging, respectively) independently reviewed all images using the PACS system, and were blinded to

**Table 1. Image parameters.**

|  | Conventional MPRAGE | Wave-CAIPI MPRAGE |
|---|---|---|
| Field of view (mm) | 227 x 250 | 256 x 256 |
| Voxel size (mm) | 0.8 x 0.8 x 1 | 1 x 1 x 1 |
| TR (ms) | 1940 | 2500 |
| TE (ms) | 3.0 | 3.1 |
| Flip Angle | 12.0 | 9.0 |
| Band width (Hz) | 240 | 240 |
| TI (ms) | 900 | 1100 |
| NEX | 1 | 1 |
| Parallel imaging method | GRAPPA | CAIPIRINHA |
| Acceleration factor (phase encoding direction) | 2 | 2 |
| Acceleration factor 3D (slice encoding direction) | - | 2 |
| Scan time (s) | 4 min 30 sec | 2 min 39 sec |

\* Abbreviations: CAIPIRINHA, controlled aliasing in parallel imaging results in higher acceleration; GRAPPA, generalized autocalibrating partially parallel acquisitions; MPRAGE, magnetization-prepared rapid gradient echo; TE, echo time; TI, inversion time; TR, repetition time; and wave-CAIPI, wave-controlled aliasing in parallel imaging.

initial diagnosis and the results of other observers and sequences. The training session involved 10 cases of conventional and Wave-CAIPI MPRAGE from patients that had been excluded from the main analysis. Each reader assessed the images for each sequence separately with a 2-week interval to prevent recall bias.

For the evaluation of enhancing intracranial lesions, we assessed conventional and Wave-CAIPI MPRAGE using the following parameters: 1) presence or absence of enhancing lesions and 2) diagnosis of enhancing lesions based on the following 5 categories: no lesion, intra-axial tumor, intra-axial non-tumorous condition, extra-axial tumor and other lesions. We also evaluated non-enhancing lesions in the brain parenchyma for 1) the presence or absence of lesions and 2) diagnosis of the lesions according to the following 5 categories: no lesion, hemorrhage (including all stages), infarction, encephalomalacic change with undetermined cause, and other lesions. If the differential diagnosis for hemorrhage, intra-axial non-tumorous condition, or intra-axial tumor is needed, precontrast MPRAGE, T2-weighted imaging, or FLAIR were used to confirm the diagnosis. In patients with multiple pathologies, up to three different categories were diagnosed in a single patient.

For the quantitative analysis, we measured the following: 1) long and short diameters of the enhancing lesions, 2) signal-to-noise ratio (SNR) at the level of pons and centrum semiovale, 3) contrast-to-noise ratio (CNR) and 4) contrast rate of enhancing lesions. We gauged each enhancing lesion in the axial image with the longest diameter and perpendicular short diameter, excluding lesions with cystic component and poorly defined margins [16]. The SNR was measured by locating the circular ROI in three areas of brain parenchyma as follows: (a) the white matter of both centrum semiovale, and (b) the center of the pons. The signal intensity and standard deviation (SD) of the brain parenchyma (ROI size $>40$ mm$^2$) were measured in the normal brain parenchyma, and the mean values were used for the centrum semiovale. The SNR was defined as 0.695 x (signal intensity) / (noise), with noise measured as the SD of the parenchyma [17]. We did not directly obtain the noise in the air because the inhomogeneous noise distribution that results from parallel imaging. Instead, we measured the SD of the brain parenchyma [18–20]. The mean value for both sides of the centrum semiovale was used for SNR$_{centrum\ semiovale}$. For enhancing lesions, CNR and contrast rate were calculated using the following formula: CNR = SNR$_{lesion}$−SNR$_{parenchyma}$ and contrast rate = ([SI$_{lesion}$−SI$_{parenchyma}$]/ SI$_{parenchyma}$} × 100 [16, 21].

For the qualitative analysis, we evaluated gray-white differentiation and the conspicuity of enhancing lesions by visual analysis. The conspicuity of enhancing lesions was rated based on the following three-level scale: 1 = a lesion whose borders were indistinguishable from the background brain; 2 = a lesion with blurry margins; and 3 = sharp lesion margins. The gray-white matter sharpness was graded using a three-level scale: 1 = indistinguishable gray-white differentiation, 2 = blurry gray-white sharpness, and 3 = well-defined gray-white differentiation [22].

We also assessed image quality using overall image quality and motion artifact. The overall image quality and motion artifact were also assessed using a five-level scale based on visual analysis: 1 = non-diagnostic image quality due to strong artifacts; 2 = severe blurring that resulted in significant limitation in evaluation; 3 = moderate blurring that slightly compromised evaluation; 4 = slight blurring that did not compromise image assessment; and 5 = excellent image quality without artifacts.

## Statistical analysis

For statistical analysis, we used MedCalc, version 15.0 (MedCalc Software, Ostend, Belgium) or SPSS software (version 20.0; SPSS, Chicago, IL). All normally distributed variables were

presented as numbers and percentages for categorical variables and means with standard deviations for continuous variables. The agreement for the presence and diagnoses of enhancing and non-enhancing lesions between both conventional and Wave-CAIPI MPRAGE were analyzed by using percent agreement and the weighted kappa (κ) value. The diameters of enhancing lesions, SNR, CNR, and contrast rate were compared using the paired T-test. The image SNR, grey-white matter, and conspicuity of enhancing lesion, overall image quality, and motion artifact were compared using the Kruskal-Wallis test. For intraobserver and interobserver agreement, weighted kappa was used for the diagnosis and presence of enhancing and non-enhancing lesions. The strength of agreement of the k-values was categorized as follows: less than 0.20, poor; 0.21–0.40, fair; 0.41–0.60, moderate; 0.61–0.80, good; and 0.81–1.00, excellent [23]. A *p*-value of less than 0.05 was considered statistically significant.

## Results

A total of 233 participants were enrolled in the retrospective study (male: female = 101:132; mean age: 58.6 ± 17.3 years [range, 21–94]).

### Diagnosis of intracranial lesions

For the detection and diagnosis of intracranial lesions, the agreement of conventional and Wave-CAIPI MPRAGE is shown in Table 1. In terms of the presence of enhancing lesions, the agreement was 98.7% (460 of 466) with excellent agreement (k = 0.965) in both sequences in the pooled analysis (Table 2). Using five categories for the diagnosis of enhancing lesions, 97.6% agreement (455 of 466, k = 0.955) was achieved from the 73 lesions of 70 patients. The diagnoses were 15 intra-axial tumors (11 metastases (Fig 1) and four other tumors), 28 extra-axial tumors (16 meningiomas (Fig 2), four schwannomas (Fig 3), four bone tumors including bone metastases and four other tumors), five intra-axial non-tumorous condition (two multiple sclerosis and three other diseases) and 25 other disease (eight developmental venous

**Table 2. Agreement of diagnosis of intracranial lesions.**

| | Percent agreement (# of agreement/total) | Kappa value/ICC (95% CI) |
|---|---|---|
| **Presence of enhancing lesions** | | |
| Overall | 98.7% (460/466) | 0.965 (0.938–0.992) |
| Observer 1 | 97.8% (228/233) | 0.942 (0.893–0.992) |
| Observer 2 | 99.5% (232/233) | 0.988 (0.966–1.000) |
| **Diagnosis of enhancing lesions** | | |
| Overall | 97.6% (455/466) | 0.955 (0.920–0.990) |
| Observer 1 | 97.8% (228/233) | 0.939 (0.891–0.987) |
| Observer 2 | 99.5% (232/233) | 0.985 (0.956–1.000) |
| **Presence of non-enhancing lesions** | | |
| Overall | 97.6% (455/466) | 0.952 (0.922–0.981) |
| Observer 1 | 97.4% (227/233) | 0.942 (0.896–0.988) |
| Observer 2 | 97.8% (228/233) | 0.953 (0.905–1.000) |
| **Diagnosis of non-enhancing lesions** | | |
| Overall | 96.9% (455/466) | 0.945 (0.913–0.978) |
| Observer 1 | 96.1% (224/233) | 0.941 (0.896–0.988) |
| Observer 2 | 97.8% (228/233) | 0.961 (0.925–0.999) |

* Abbreviation: CI, confidence interval.

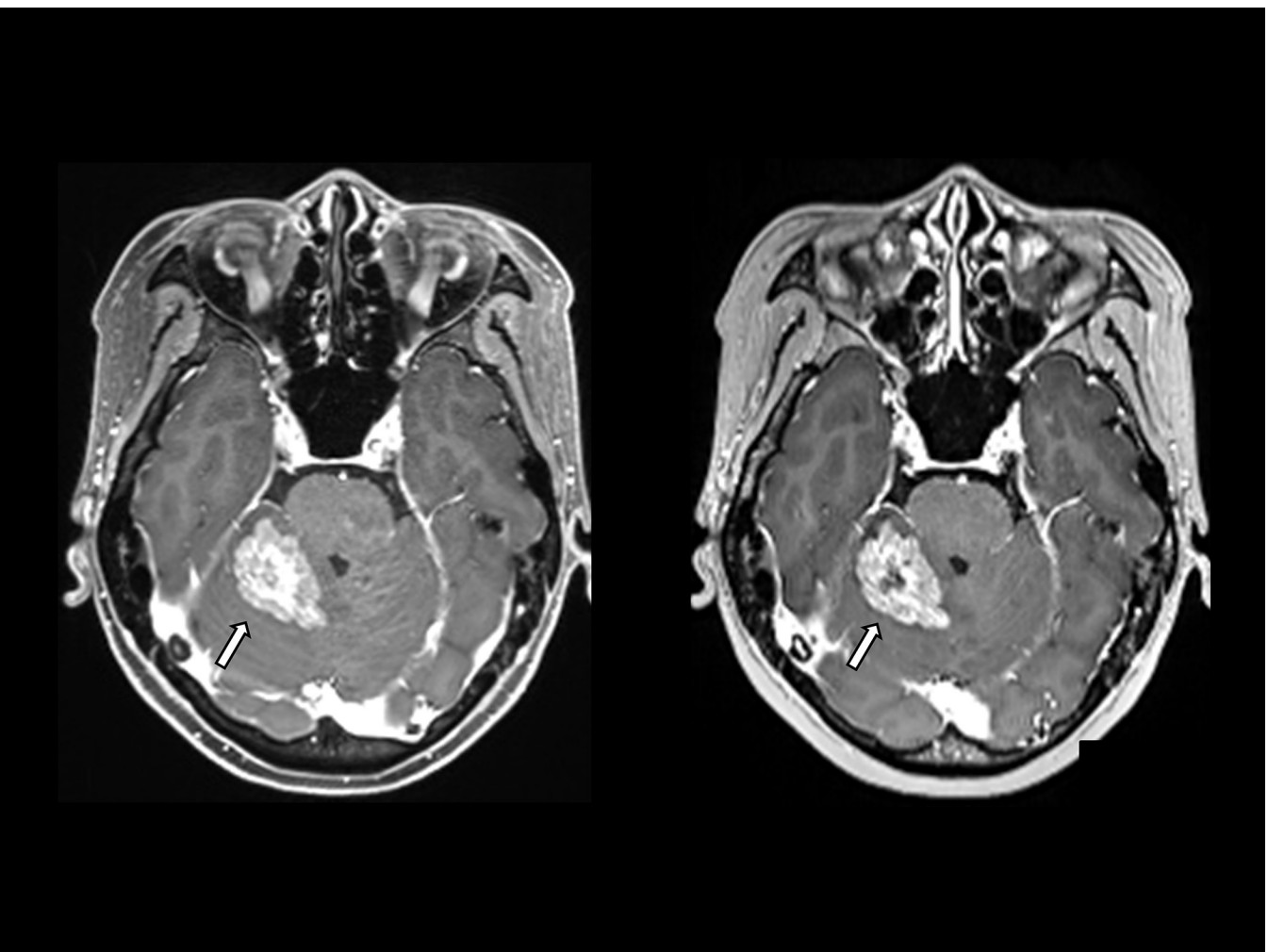

**Fig 1. A 49-year-old woman with metastasis on conventional and Wave-CAIPI MPRAGE.** A 49-year-old woman with breast cancer presented approximately 3.5cm enhancing lesion with perilesional edema in the right superior cerebellum, causing mild compression of the right side of the 4th ventricle, suggestive of metastasis. It is well demonstrated on both enhanced conventional MPRAGE (left) and Wave-CAPI MPRAGE (right).

anomalies, four spontaneous intracranial hypotension, three meningitis, one arteriovenous malformation, and seven other diseases).

Intraobserver agreement for the presence and diagnosis of enhancing lesions were excellent in both sequences (conventional MPRAGE vs. Wave-CAIPI MPRAGE [k, confidential interval] = 0.842 [0.768–0.916]: 0.834 [0.759–0.909] for the presence of enhancing lesions; and 0.803 [0.707–0.899]: 0.822 [0.731–0.914] for diagnosis of enhancing lesions). Interobserver agreement for the presence and diagnosis of enhancing lesions were excellent to good in both conventional MPRAGE (conventional MPRAGE vs. Wave-CAIPI MPRAGE [k, confidential interval] = 0.841 [0.761–0.922]: 0.839 [0.758–0.921] for the presence of enhancing lesions; and 0.737 [0.635–0.840]: 0.747 [0.647–0.847] for diagnosis of enhancing lesions).

For the presence or absence of non-enhancing lesions, the conventional and Wave-CAIPI MPRAGE demonstrated 97.6% agreement (455 of 466) with excellent agreement (k = 0.952, Table 2). In total, 103 non-enhancing lesions from 77 patients were diagnosed in both sequences. The diagnoses included 23 hemorrhages (13 intracranial hemorrhages, three

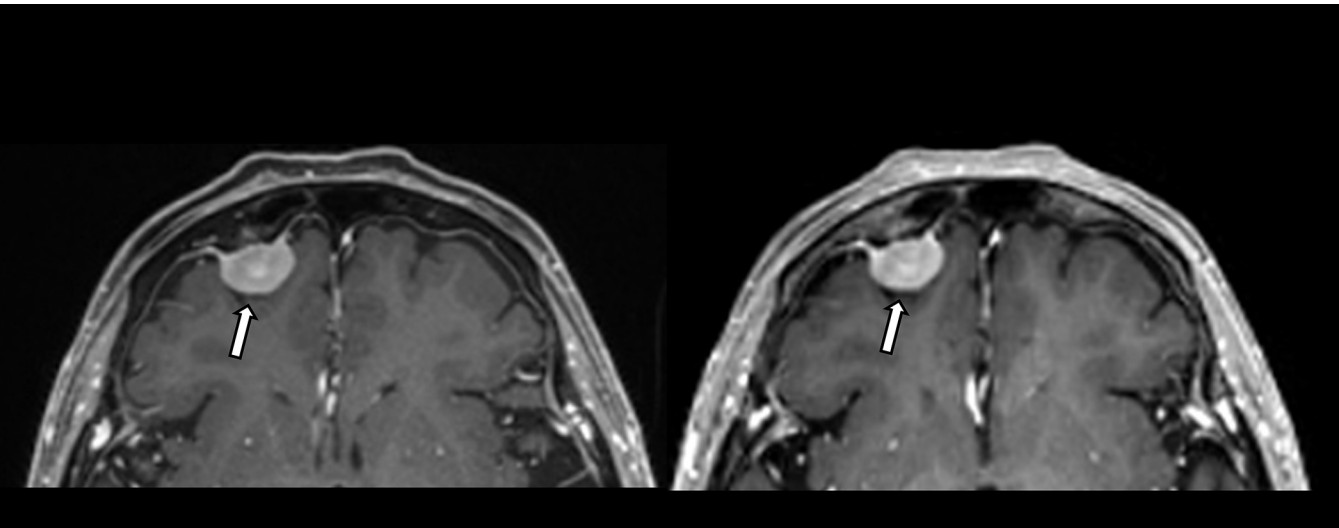

**Fig 2. A 91-year-old woman with meningioma on conventional and Wave-CAIPI MPRAGE.** A 91-year-old woman had an approximately 1.6-cm enhancing extra-axial mass in the right frontal region with a dura tail, suggestive of meningioma. It is well demonstrated on both enhanced conventional MPRAGE (left) and Wave-CAPI MPRAGE (right).

subdural hemorrhages, and two subarachnoid hemorrhages), 51 infarctions (31 lacunar infarctions and 20 territorial infarctions), 19 encephalomalatic change, and 10 other diseases.

## Quantitative and qualitative analysis of image parameters

We performed both quantitative (Table 3) and qualitative analysis (Table 4) to compare the image parameters of conventional and Wave-CAIPI MPRAGE. In quantitative analysis, the long and short diameters of the enhancing lesion did not differ between the two sequences

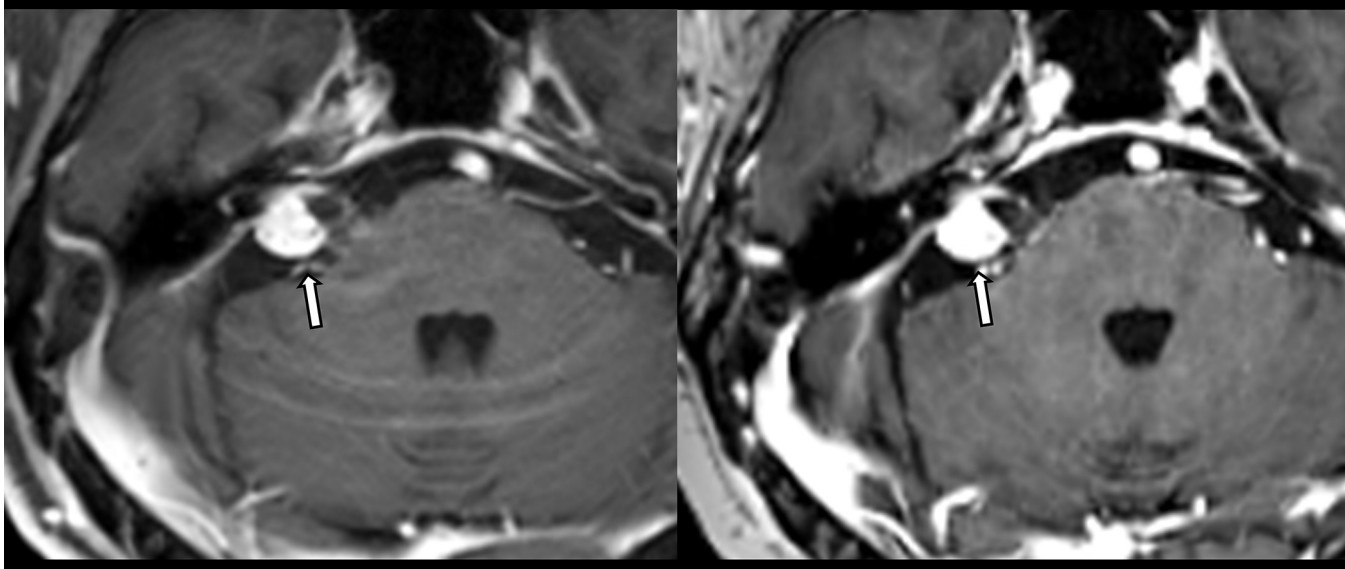

**Fig 3. An 84-year-old man with vestibular schwannoma on conventional and Wave-CAIPI MPRAGE.** An 84-year-old man presented with dizziness. Approximately 1.6-cm cm enhancing extra-axial mass in the right internal auditory canal causes internal auditory canal widening, suggestive of vestibular schwannoma. It is easily visualized on both enhanced conventional MPRAGE (left) and Wave-CAPI MPRAGE (right).

**Table 3. Quantitative analysis of imaging parameters between conventional and Wave-CAIPI MPRAGE.**

| | Conventional MPRAGE (Mean ± SD [Range], mm) | Wave-CAIPI MPRAGE (Mean ± SD [Range], mm) | P-value |
|---|---|---|---|
| **Long diameter** | | | |
| Overall | 16.90 ± 11.42 (2–60) | 16.83 ± 11.34 (2–63) | 0.615 |
| Observer 1 | 17.89 ± 11.36 (2–60) | 18.04 ± 11.54 (2–63) | 0.527 |
| Observer 2 | 16.26 ± 11.56 (2–58) | 16.05 ± 11.27 (2–57) | 0.246 |
| **Short diameter** | | | |
| Overall | 11.71 ± 8.52 (2–43) | 11.86 ± 8.47 (2–42) | 0.159 |
| Observer 1 | 12.48 ± 8.59 (2–43) | 12.48 ± 8.59 (2–42) | 0.602 |
| Observer 2 | 11.21 ± 8.54 (2–36) | 11.40 ± 8.69 (2–39) | 0.186 |
| **$SNR_{centrum\ semiovale}$** | | | |
| Overall | 51.35 ± 16.61 (49.74–52.85) | 40.00 ± 11.55 (38.99–41.07) | 0.001 |
| Observer 1 | 51.89± 16.92 (49.65–54.09) | 40.42 ± 11.67 (38.92–41.98) | 0.001 |
| Observer 2 | 50.82 ±16.30 (48.65–52.86) | 39.59 ± 11.44 (37.99–40.98) | 0.001 |
| **$SNR_{pons}$** | | | |
| Overall | 60.38 ± 21.87 (58.28–62.29) | 46.73 ± 13.13 (45.59–47.92) | 0.001 |
| Observer 1 | 60.14 ± 21.97 (57.15–63.02) | 46.67± 13.18 (45.01–48.43) | 0.001 |
| Observer 2 | 60.62± 21.80 (57.86–63.36) | 46.79 ± 13.11 (44.96–48.45) | 0.001 |
| **CNR** | | | |
| Overall | 28.14 ± 15.67 (7.07–87.42) | 26.56 ± 17.47 (2.99–82.44) | 0.486 |
| Observer 1 | 28.08 ± 18.69 (7.89–87.42) | 25.61 ± 15.15 (5.31–77.70) | 0.405 |
| Observer 2 | 28.20 ± 13.10 (7.07–59.42) | 27.31 ± 19.30 (2.99–82.44) | 0.795 |
| **Contrast Rate** | | | |
| Overall | 72.70 ± 29.15 (15.56–151.63) | 88.94 ± 48.65 (18.37–218.46) | <0.001 |
| Observer 1 | 70.57 ± 30.47 (15.56–133.63) | 89.67 ± 42.88 (22.20–174.43) | <0.001 |
| Observer 2 | 74.40 ± 28.41 (22.04–151.63) | 88.35 ± 53.43 (18.37–218.46) | 0.029 |

* Abbreviations: CNR, contrast-to-noise ratio; MPRAGE, magnetization-prepared rapid gradient echo; SD, standard deviation; SNR, signal-to-noise ratio; and wave-CAIPI, wave-controlled aliasing in parallel imaging.

**Table 4. Qualitative analysis.**

| | Conventional MPRAGE (Mean ± SD, Median (interquartile range; range)) | Wave-CAIPI MPRAGE (Mean ± SD, Median (interquartile range; range)) | P-value |
|---|---|---|---|
| **Gray-White Differentiation** | | | |
| Overall | 2.95 ± 0.31, 3 (3–3; 1–3) | 2.97 ± 0.21, 3 (3–3; 1–3) | 0.375 |
| Observer 1 | 2.88 ± 0.36, 3 (3–3; 1–3) | 2.93 ± 0.29, 3 (3–3; 1–3) | 0.028 |
| Observer 2 | 2.96 ± 0.23, 3 (3–3; 1–3) | 3.00 ± 0.01, 3 (3–3; 2–3) | 0.083 |
| **Conspicuity of enhancing lesion** | | | |
| Overall | 2.84 ± 0.40, 3 (3–3; 1–3) | 2.93 ± 0.25, 3 (3–3; 1–3) | 0.109 |
| Observer 1 | 2.96 ± 0.20, 3 (3–3; 2–3) | 2.91 ± 0.29, 3 (3–3; 2–3) | 0.564 |
| Observer 2 | 2.80 ± 0.45, 3 (3–3; 1–3) | 2.94 ± 0.23, 3 (3–3; 2–3) | 0.035 |

* Abbreviations: MPRAGE, magnetization-prepared rapid gradient echo; SD, standard deviation; and wave-CAIPI, wave-controlled aliasing in parallel imaging.

**Table 5. Comparisons of image qualities.**

| | Conventional MPRAGE (Mean ± SD, Median [interqultile range; range]) | Wave-CAIPI MPRAGE (Mean ± SD, Median [interqultile range; range]) | P-value |
|---|---|---|---|
| **Overall image quality** | | | |
| Overall | 4.49 ± 0.42, 4 (4–5; 2–5) | 4.46 ± 0.60, 4 (4–5; 2–5) | 0.005 |
| Observer 1 | 4.23 ± 0.59, 4 (4–5; 2–5) | 4.41 ± 0.65, 4 (4–5; 2–5) | <0.001 |
| Observer 2 | 4.75 ± 0.36, 4 (4–5; 3–5) | 4.51 ± 0.55, 5 (5–5; 3–5) | 0.533 |
| **Motion artifact** | | | |
| Overall | 4.77 ± 0.55, 5 (5–5; 2–5) | 4.84 ± 0.48, 5 (5–5; 2–5) | 0.005 |
| Observer 1 | 4.86 ± 0.63, 4 (4–5; 2–5) | 4.75 ± 0.59, 4 (4–5; 2–5) | 0.102 |
| Observer 2 | 4.86 ± 0.43, 4 (4–5; 3–5) | 4.93 ± 0.32, 5 (5–5; 3–5) | 0.002 |

* Abbreviations: MPRAGE, magnetization-prepared rapid gradient echo; SD, standard deviation; and wave-CAIPI, wave-controlled aliasing in parallel imaging.

(p > 0.05 for all). The SNR at both the level of the centrum semiovale and pons was reduced in Wave-CAIPI MPRAGE (both p < 0.05). However, CNR of enhancing lesions did not differ between the two sequences (p > 0.05), and the contrast rate was significantly higher than that of conventional images (p < 0.001). In qualitative analysis, the conspicuity of enhancing lesions and gray-white matter differentiation showed no difference between the two sequences (both p > 0.05).

## Comparison of image qualities

The overall image quality of Wave-CAIPI MPRAGE was slightly poorer than that of conventional images (p = 0.005, Table 5). However, both sequences achieved a median value of score 4 (indicating slight blurring that did not compromise image assessment) and motion artifacts is better in Wave-CAIPI MPRAGE (p = 0.005, Table 5).

## Discussion

In our study, Wave-CAIPI MPRAGE was found to be comparable to conventional MPRAGE for the diagnosis of enhancing intracranial lesions, as well as non-enhancing intracranial lesions, with 2 min 30 sec of the acquisition time. Wave-CAIPI MPRAGE also showed a higher contrast rate and similar CNR compared with those of conventional MPRAGE. Moreover, the diameter and conspicuity of enhancing lesions and gray-white matter differentiation were also similar in both sequences. Even though the SNR and overall image quality were slightly poorer in Wave-CAIPI MPRAGE, both Wave-CAIPI MPRAGE and conventional MPRAGE achieved a median value of score 4 which suggests preservation of appropriate overall image quality without compromising image assessment. Considering those results, we concluded that post-contrast Wave-CAIPI MPRAGE could be a viable option of faster MR acquisition for the diagnosis of intracranial lesions in our clinical practice.

For the evaluation of enhancing intracranial lesions including metastasis and glioma, high spatial resolution images such as 1 mm isovoxel scan of MPRAGE are recommended to prevent misdiagnosis of tiny lesions by guidelines [4, 24]. Therefore, there has been continuous effort toward technical advances in MR imaging to fulfill both high spatial resolution and sufficient image quality within an acceptable scan time. Wave-CAIPI MPRAGE was developed to perform a highly accelerated scan with reduced scan time and preserved image quality

compared with conventional MRPAGE [10, 25]. The Wave-CAIPI technique prevents the increase of g-factor from the summation of aliased voxels of highly accelerated images, which amplifies the noise and eventually lowers the diagnostic accuracy and image quality, by using the characteristic corkscrew k-space trajectory [10, 25, 26]. Previous clinical study reported precontrast Wave-CAIPI MRPAGE could have enough spatial resolution for the evaluation of dementia and high scan-rescan reliability [12, 13]. One step further, here we suggest that Wave-CAIPI MPRAGE based highly accelerated MRI could be a reliable method for the diagnosis of enhancing intracranial lesions.

To date, Wave-CAIPI MPRAGE presents some challenges that need to be overcome. A previous study reported that preconstrast Wave-CAIPI MPRAGE images showed more image noise than conventional MPRAGE, especially in the central brain due to the following reasons: 1) a relatively lower SNR in the center of the coil in comparison to the periphery, and 2) reduction in the SNR with the square root of the acceleration factor [13, 25]. Although previous technical developments have been implemented to minimize both noise amplification and wave-specific blurring artifacts [12], we also observed a lower SNR in Wave-CAIPI MPRAGE than in conventional MPRAGE. However, Wave-CAIPI MPRAGE demonstrated considerably high agreement with conventional MPRAGE in terms of volumetric analysis and visual grading of parenchymal atrophy [12, 13]. Also, there have been approaches to apply Wave-CAIPI to other sequences such as FLAIR or 3D fast/turbo spine echo image post-contrast T1 (SPACE) in diagnosing intracranial lesions. One study comparing the cerebral white matter lesion volume between Wave-FLAIR and conventional FLAIR showed comparable diagnostic quality [27]. Further, recent studies applying Wave-CAIPI to post-contrast T1 (SPACE) revealed that fast scan using Wave-CAIPI provided equivalent visualization of enhancing lesions and overall diagnostic quality for evaluating intracranial enhancing lesions [28, 29].

We also noted that Wave-CAIPI MPRAGE achieved a high agreement with conventional MPRAGE for diagnosis, measurement of diameter, and conspicuity of enhancing lesions. Considering these evidences, we assumed that a mild decreased SNR in Wave-CAIPI MPRAGE could have a negligible impact on the clinical decision by radiologists in daily practice.

In the aspect of image quality, the overall image quality was reduced in Wave-CAIPI MPRAGE. Similar to the SNR, three times higher acceleration factor of Wave-CAIPI could affect noise amplification and result in lower overall image quality. However, the decreased overall image quality of Wave-CAIPI MPRAGE was slight blurring of images that did not compromise image assessment. Therefore, we also consider that the diagnostic performance of Wave-CAIPI will not be affected. In addition, a previous technical study raised the possibility of greater motion artifact of the images using Wave-CAIPI technique [10]. In contrast, similar to results obtained in a clinical study using Wave-CAIPI SWI, the motion artifact of Wave-CAIPI MPRAGE was less pronounced than that of conventional MRPRAGE in this study [30, 31]. We anticipated that a shorter scan time of Wave-CAIPI MPRAGE resulted in fewer chances to meet patient motion during MR imaging. Moreover, further improvement of post-processing techniques of Wave-CAIPI MPRAGE, such as denoising and image regularization, could be helpful to decrease the noise in the Wave-CAIPI MPRAGE images without causing excessive spatial blurring.

This study has several limitations to note. First, the participants were recruited from a single center and a relatively small number of enhancing lesions were included. However, determination of the presence and absence of the lesion could be an essential first step for the validation of new sequences for clinical application; therefore, we did not restrict the inclusion criteria for the patients with enhancing intracranial lesions for proper validation of Wave-CAIPI MPRAGE. Second, the diagnosis of intracranial lesions was based on the findings of MR

imaging, since histopathologic confirmation was difficult for intracranial lesions. Thus, we focused on the evaluation of diagnostic agreement at a single point of MR examination rather than a diagnostic accuracy test based on pathologic confirmation. Additionally, various types of presumed diagnoses were evaluated in this study. We designed this study for the initial validation of new sequences by testing the presence and absence of lesions. Based on our results, attempts to further studies about the variable clinical applications of metastasis and the optimization of scan parameters in each sequence of Wave-CAIPI MPRAGE may be warranted. Lastly, even though the entire study sample showed the equivalent conspicuity of enhancing lesions in this study, image scanning order might have been influenced conspicuity of enhancement due to the time elapsed from contrast injection. It seems alternating the acquisition order and comparing the images with different acquisition order is optimal to prevent the possible bias in determining conspicuity of enhancement, but it was not feasible in daily practice since it could cause other bias during controlling the sequence or communication error among the technologists. Alternating the acquisition order in scan-by scan manner could be a possible solution to resolve the issue in the future study. In conclusion, Wave-CAIPI MPRAGE provide a reliable diagnostic agreement for the diagnosis of enhancing intracranial lesions compared with conventional MPRAGE with almost half of the scan time. Wave-CAIPI MPRAGE also achieved high agreement for the diagnosis of non-enhancing lesions and measurement of enhancing lesions compared to conventional MPRAGE. Considering the decreased scan time and similar diagnostic performance, Wave-CAIPI MRPAGE could be an efficient alternative for MR imaging in our daily practice.

## Supporting information

**S1 File.**
(XLSX)

## Acknowledgments

The Wave-CAIPI sequence was obtained from Dr. Kawin Setsompop and Dr. Stephen Cauley at the A.A. Martinos center, through the Siemens Work In Progress sharing program. InSeong Kim at Siemens Healthineers Ltd. helped to install and optimize scan parameters of Wave-CAIPI MPRAGE.

## Author Contributions

**Conceptualization:** Hyunji Oh, Younghee Yim, Mi Sun Chung, Jun Soo Byun.

**Data curation:** Hyunji Oh, Younghee Yim, Mi Sun Chung, Jun Soo Byun.

**Formal analysis:** Hyunji Oh, Younghee Yim, Mi Sun Chung.

**Funding acquisition:** Hyunji Oh.

**Methodology:** Hyunji Oh.

**Project administration:** Younghee Yim.

**Resources:** Younghee Yim.

**Supervision:** Younghee Yim.

**Visualization:** Mi Sun Chung, Jun Soo Byun.

**Writing – original draft:** Hyunji Oh, Younghee Yim, Mi Sun Chung, Jun Soo Byun.

**Writing – review & editing:** Hyunji Oh, Younghee Yim, Mi Sun Chung, Jun Soo Byun.

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
