## [Decision Letter · Decision Letter 0]

9 Mar 2023

PONE-D-21-38646Wave-controlled aliasing in parallel imaging (Wave-CAIPI): Accelerating speed for the MRI-based diagnosis of enhancing intracranial lesions compared to magnetization-prepared gradient echoPLOS ONE

Dear Dr. Yim,

Thank you for submitting your manuscript to PLOS ONE. After careful consideration, we feel that it has merit but does not fully meet PLOS ONE’s publication criteria as it currently stands. Therefore, we invite you to submit a revised version of the manuscript that addresses the points raised during the review process.

We look forward to receiving your revised manuscript.

Kind regards,

Cem M. Deniz

Academic Editor

PLOS ONE

Journal Requirements:

“YY got grant for this research.

This research was supported by the Chung-Ang University Research Grants in 2021.”

Additional Editor Comments (if provided):

Your manuscript was reviewed by an expert in diagnostic radiology. Reviewer found the study to be a complementary to previous research and it has a broad interest. I agree with the reviewer and suggest a minor revision for the manuscript.

Reviewers' comments:

Reviewer's Responses to Questions

**Comments to the Author**

1. Is the manuscript technically sound, and do the data support the conclusions?

Reviewer #1: Yes

2. Has the statistical analysis been performed appropriately and rigorously? 

Reviewer #1: Yes

3. Have the authors made all data underlying the findings in their manuscript fully available?

Reviewer #1: Yes

4. Is the manuscript presented in an intelligible fashion and written in standard English?

Reviewer #1: Yes

5. Review Comments to the Author

Reviewer #1: This is a well-conducted study evaluating an ultrafast MRI technique, wave-CAIPI, against conventional post-contrast imaging using the MPRAGE sequence. The authors demonstrate high agreement for the detection and diagnosis of enhancing lesions using wave-CAIPI MPRAGE as compared to conventional MPRAGE in half the scan time. The manuscript is clearly written, and the text is supported by appropriately chosen tables and figures. This work complements recently published work on the clinical validation of the wave-CAIPI technique and will be of broad interest to clinicians and researchers in neuroimaging.

Abstract: The motivation and purpose are clearly defined, and the conclusion is consistent with the findings of the paper.

Introduction: The introduction provides a good overview of the problem and provides appropriate motivation and context for the current comparative evaluation of post-contrast wave-CAIPI MPRAGE. It would be helpful to cite recently published work evaluating post-contrast wave-CAIPI MPRAGE for detection of enhancing lesions, which the results of the current manuscript affirm: Filho ALMG et al. Eur Radiol. 2022 Dec 2;1-11.

Methods:

- Image Acquisition: Was the order of wave-CAIPI and conventional MPRAGE imaging randomized after contrast injection? The time elapsed between injection of contrast and imaging may influence the conspicuity of enhancing lesions, particularly on the MPRAGE sequence. A sentence clarifying the order of the sequences should be included in the methods, and a discussion of the order and its impact on the results should be included in the discussion.

Results:

- Were any other artifacts other than motion (e.g., flow-related artifact) evaluated for? Could these artifacts contribute to false positives for enhancing lesions?

- Otherwise, thoroughly described and transparently documented in the tables and figures.

Discussion:

- The authors should provide a comprehensive discussion of other wave-CAIPI papers that have been published demonstrating the equivalent evaluation of enhancing lesions using wave-CAIPI T1 SPACE (see: Filho ALMG et al. J Neuroimaging. 2021 Sep;31(5):893-901; Filho ALMG et al. Front Neurol. 2020 Oct 27;11:587327) and the evaluation of demyelinating and white matter lesions using wave-CAIPI FLAIR imaging (see: Ngamsombat C et al. AJNR Am J Neuroradiol. 2021 Sep;42(9):1584-1590).

Conclusion:

- Concise and highlights the main conclusions of this work.

References: Aside from the additions suggested above, appropriate as cited.

6. PLOS authors have the option to publish the peer review history of their article (what does this mean?). If published, this will include your full peer review and any attached files.

Reviewer #1: No

---

## [Author Response · Author response to Decision Letter 0]

22 Mar 2023

First of all, your requests and comments not only strengthened our article but also widened our perspectives. We tried our best to embrace your remarks on the submission web site as much as maintaining the original contents. Below are responses and answers referring your notes; we are always ready to take your further requests and constructive opinions if needed. We appreciate your time and efforts.

Reviewer #1: This is a well-conducted study evaluating an ultrafast MRI technique, wave-CAIPI, against conventional post-contrast imaging using the MPRAGE sequence. The authors demonstrate high agreement for the detection and diagnosis of enhancing lesions using wave-CAIPI MPRAGE as compared to conventional MPRAGE in half the scan time. The manuscript is clearly written, and the text is supported by appropriately chosen tables and figures. This work complements recently published work on the clinical validation of the wave-CAIPI technique and will be of broad interest to clinicians and researchers in neuroimaging.

Abstract: The motivation and purpose are clearly defined, and the conclusion is consistent with the findings of the paper.

Introduction: The introduction provides a good overview of the problem and provides appropriate motivation and context for the current comparative evaluation of post-contrast wave-CAIPI MPRAGE. It would be helpful to cite recently published work evaluating post-contrast wave-CAIPI MPRAGE for detection of enhancing lesions, which the results of the current manuscript affirm: Filho ALMG et al. Eur Radiol. 2022 Dec 2;1-11.

Response to the comment: Thank you for your valuable suggestions. We added the recent research as additional reference as you suggested. 

Also, recent study demonstrated that fast scan using contrast-enhanced Wave-CAIPI 3D T1-MPRAGE was noninferior to the 3D T1-MPRAGE sequence in visualizing and diagnosing enhancing brain lesions.

Methods:

- Image Acquisition: Was the order of wave-CAIPI and conventional MPRAGE imaging randomized after contrast injection? The time elapsed between injection of contrast and imaging may influence the conspicuity of enhancing lesions, particularly on the MPRAGE sequence. A sentence clarifying the order of the sequences should be included in the methods, and a discussion of the order and its impact on the results should be included in the discussion.

Response to the comment: Thank you for your valuable suggestions. We added related sentences as below.

Methods;

Post-contrast MR scanning was executed just after the injection of contrast media in the following order: Wave-CAIPI MPRAGE � conventional MPRAGE.

Discussion;

Lastly, even though the entire study sample showed the equivalent conspicuity of enhancing lesions in this study, image scanning order might have been influenced conspicuity of enhancement due to the time elapsed from contrast injection. It seems alternating the acquisition order and comparing the images with different acquisition order is optimal to prevent the possible bias in determining conspicuity of enhancement, but it was not feasible in daily practice since it could cause other bias during controlling the sequence or communication error among the technologists. Alternating the acquisition order in scan-by scan manner could be a possible solution to resolve the issue in the future study.

Results:

- Were any other artifacts other than motion (e.g., flow-related artifact) evaluated for? Could these artifacts contribute to false positives for enhancing lesions?

- Otherwise, thoroughly described and transparently documented in the tables and figures.

Response to the comment: Thank you for your valuable comment.

We agree with your idea that artifacts can affect creating pseudo-lesions. However, observers did not include artifact-like pseudo-lesions in their diagnosis as we underwent training session before the analysis and discussed what to diagnose as true lesion. Also, we excluded all images with severe artifact that could create false positive lesions in this study.

Discussion:

- The authors should provide a comprehensive discussion of other wave-CAIPI papers that have been published demonstrating the equivalent evaluation of enhancing lesions using wave-CAIPI T1 SPACE (see: Filho ALMG et al. J Neuroimaging. 2021 Sep;31(5):893-901; Filho ALMG et al. Front Neurol. 2020 Oct 27;11:587327) and the evaluation of demyelinating and white matter lesions using wave-CAIPI FLAIR imaging (see: Ngamsombat C et al. AJNR Am J Neuroradiol. 2021 Sep;42(9):1584-1590).

Response to the comment: Thank you for your valuable suggestions. We included related contents in discussion section as below.

Also, there have been approaches to apply Wave-CAIPI to other sequences such as FLAIR or 3D fast/turbo spine echo image post-contrast T1 (SPACE) in diagnosing intracranial lesions. One study comparing the cerebral white matter lesion volume between Wave-FLAIR and conventional FLAIR showed comparable diagnostic quality [27]. Further, recent studies applying Wave-CAIPI to post-contrast T1 (SPACE) revealed that fast scan using Wave-CAIPI provided equivalent visualization of enhancing lesions and overall diagnostic quality for evaluating intracranial enhancing lesions [28, 29].

27. American Journal of Neuroradiology September 2021, 42 (9) 1584-1590;

Evaluation of Ultrafast Wave–Controlled Aliasing in Parallel Imaging 3D-FLAIR in the Visualization and Volumetric Estimation of Cerebral White Matter Lesions

28. Front Neurol. 2020 Oct 27;11:587327. doi: 10.3389/fneur.2020.587327. eCollection 2020.

Accelerated Post-contrast Wave-CAIPI T1 SPACE Achieves Equivalent Diagnostic Performance Compared With Standard T1 SPACE for the Detection of Brain Metastases in Clinical 3T MRI

29. J Neuroimaging. 2021 Sep;31(5):893-901. doi: 10.1111/jon.12893. Epub 2021 Jun 3.

MRI Highly Accelerated Wave-CAIPI T1-SPACE versus Standard T1-SPACE to detect brain gadolinium-enhancing lesions at 3T

Conclusion:

- Concise and highlights the main conclusions of this work.

References: Aside from the additions suggested above, appropriate as cited.

---

## [Decision Letter · Decision Letter 1]

17 Apr 2023

Wave-controlled aliasing in parallel imaging (Wave-CAIPI) : Accelerati ng speed for the MRI-based diagnosis of enhancing intracranial lesions compared to magnetization-prepared gradient echo

PONE-D-21-38646R1

Dear Dr. Yim,

We’re pleased to inform you that your manuscript has been judged scientifically suitable for publication and will be formally accepted for publication once it meets all outstanding technical requirements.

Kind regards,

Cem M. Deniz

Academic Editor

PLOS ONE

Additional Editor Comments (optional):

Congratulations on the acceptance of your manuscript.

Reviewers' comments:

Reviewer's Responses to Questions

**Comments to the Author**

1. If the authors have adequately addressed your comments raised in a previous round of review and you feel that this manuscript is now acceptable for publication, you may indicate that here to bypass the “Comments to the Author” section, enter your conflict of interest statement in the “Confidential to Editor” section, and submit your "Accept" recommendation.

Reviewer #1: All comments have been addressed

2. Is the manuscript technically sound, and do the data support the conclusions?

Reviewer #1: Yes

3. Has the statistical analysis been performed appropriately and rigorously? 

Reviewer #1: Yes

4. Have the authors made all data underlying the findings in their manuscript fully available?

Reviewer #1: Yes

5. Is the manuscript presented in an intelligible fashion and written in standard English?

Reviewer #1: Yes

6. Review Comments to the Author

Reviewer #1: The authors have thoroughly addressed all of my review comments. I have no further edits to suggest. I commend the authors on a job well done.

7. PLOS authors have the option to publish the peer review history of their article (what does this mean?). If published, this will include your full peer review and any attached files.

Reviewer #1: No

---

## [Editor Report · Acceptance letter]

27 Apr 2023

PONE-D-21-38646R1 

Wave-controlled aliasing in parallel imaging (Wave-CAIPI): Accelerating speed for the MRI-based diagnosis of enhancing intracranial lesions compared to magnetization-prepared gradient echo 

Dear Dr. Yim:

I'm pleased to inform you that your manuscript has been deemed suitable for publication in PLOS ONE. Congratulations! Your manuscript is now with our production department. 

Kind regards, 

on behalf of

Dr. Cem M. Deniz 

Academic Editor

PLOS ONE